# Errors in Implant Positioning Due to Lack of Planning: A Clinical Case Report of New Prosthetic Materials and Solutions

**DOI:** 10.3390/ma13081883

**Published:** 2020-04-16

**Authors:** Marco Tallarico, Roberto Scrascia, Marco Annucci, Silvio Mario Meloni, Aurea Immacolata Lumbau, Alba Koshovari, Erta Xhanari, Matteo Martinolli

**Affiliations:** 1Department of Periodontology and Implantology, University of Sassari, 07021 Sassari, Italy; me@studiomarcotallarico.it (M.T.); alumbau@uniss.it (A.I.L.); 2Private Practice, 74121 Taranto, Italy; roberto.scrascia@gmail.com; 3Private Practice, 00100 Rome, Italy; dentalcaddesign@gmail.com; 4Department of Implantology and Prosthetic Aspects, Aldent University, 1022 Tirana, Albania; alba.koshovari@ual.edu.al (A.K.); info@ertaxhanari.com (E.X.); 5Private Practice, 45014 Porto Viro, Italy; matteo.martinolli@hotmail.it

**Keywords:** dental implants, abutments, screw-retained restoration, CAD/CAM, guided surgery

## Abstract

The achievement of the optimal implant position is a critical consideration in implant surgery, as it can facilitate the ideal prosthesis design and allow adequate oral hygiene maintenance. The switch from bone-driven to prosthetic-driven implant placement, through a comprehensive diagnosis and adequate treatment plan, is a prerequisite for long-term successful implant-based therapy. The aim of the present case report is to describe a step-by-step prosthetic retreatment of a patient with primary treatment failure due to incorrect dental implant placement. Although dental implants achieve high survival rates, the success of implant prosthetic therapy significantly relies on an appropriate implant position. Malpositioned implants can cause damage to vital structures, like nerves or vessels. Moreover, improper implant positioning can result in esthetic, biological, and technical complications and can, in extreme situations, render the desired prosthetic rehabilitation impossible to achieve.

## 1. Introduction

Integrated treatment planning with dental implants is a well-established option [1,2]. Nevertheless, the achievement of the optimal implant position, based on the prosthetic plan, is still a critical consideration in implant-based surgery [1]. An ideal prosthesis design may reduce the risk of technical and biological complicationsand allow for adequate oral hygeine maintenance [1]. Moreover, an accurate restorative-driven implant placement offers important long-term advantages, allowing for favorable esthetics and function as well as optimal occlusion and masticatory forces distribution [3]. The switch from bone-driven to prosthetic-driven implant placement through a comprehensive diagnosis and adequate treatment plan is, therefore, a prerequisite for long-term successful implant-based therapy [4].

In the conventional protocol, titanium-based implants are usually placed freehand after implant sites preparation [2]. Sand-blasted and acid-etched titanium implants are the gold standard in oral implantology, even if the search on alternative materials and/or surface treatments is growing. The aims of this research are to improve esthetics (ceramic implants) and osseointegration (hydroxyapatite coating or, recently, hydrophilic surfaces).

During surgery, the surgeon defines the implant position based on the pre-surgical evaluation and the intra-surgical findings, such as the availability of bone, position of adjacent teeth, esthetics, etc. Additional information can be obtained by using conventional radiographs and a prosthetic stent [5]. However, surgeons must demonstrate surgical and prosthetic skills. Static computer-assisted template-based implant surgery involves virtual planning of the implant placement in the optimal restorative position and then utilizes surgical guides to help the surgeon perform the osteotomy and site preparation in an accurate and efficient manner [6,7,8,9].

Developments of new technologies and advanced digital optical imaging have improved the predictability, safety, and efficiency of prosthetically-driven implant placement by using computer-guided surgery [10,11,12,13,14]. However, instead, the inability to achieve an accurate diagnosis and to predict realistic outcomes of treatment may lead clinicians to several problems during the execution of treatment, increasing the risk of biological and technical complications, which can result in a treatment failure [10]. Among these, there are some mechanical drawbacks related to the lack of prosthetic space or wrong prosthetic design, such us phonetic problems and the inability to maintain oral hygiene at home, as well as esthetic problems. These represent the most common issues [4]. Early detection of potential errors, prior to the patient treatment, through a pre-visualization of the final prosthetic volumes, may contribute to the long-term success of the implant-based treatment and improved patient satisfaction [15]. The present case report describes a prosthetic retreatment of a patient with primary treatment failure due to dental implants malpositioning. The decision to remove or maintain the implants was the critical point, however, new materials and technologies have allowed a minimally invasive prosthetic retreatment.

## 2. Case Report

The objective of this case report is to present a complicated prosthetic retreatment of a primary failure, due to implant malpositioning, allowed through the use of low profile intermediate abutment, with the so-called elastic Seeger System, in combination with new digital technologies. The surgical procedures were performed by an expert clinician (MT) certified in implant-based therapy and the computer-aided design/computer-aided manufacturing (CAD/CAM) procedures by a certified expert dental technician (M.A.). The patient was informed about the nature of the study and gave their written consent for the prosthetic procedures and for the use of radiologic and clinical data for publication.

A partially edentulous 78-year-old woman with a complete screw-retained implant-support hybrid prosthesis (Figure 1) in the upper jaw and natural dentition in the lower jaw was referred to a private center in Rome, Italy, due to several continuous breakages of the prosthetic part of the implant-supported rehabilitation. After a preliminary interview, the patient stated that the implant treatment was finished two years before, but she had never been comfortable with this prosthesis (Figure 2). Relevant symptoms were phonetic difficulties, inability to maintain hygiene, and repeated breakages, leading to functional and esthetic issues (Figure 3). After that, the patient’s medical history was collected, and preoperative photographs, radiographs, periodontal screening results, and model casts were obtained for initial evaluation (Figure 4). During the clinical examination, the actual prosthesis was unscrewed, due to teeth detachments, and replaced with the old temporary prosthesis delivered by the patient. Both existing prostheses were evaluated and judged inaccurate, with particular attention to the fit of the prosthesis, the vertical dimension of occlusion, phonetics, facial support, and lip position. All the possible treatment options were then discussed and evaluated together with the patient. The main concerns of maintaining the previously placed implants were some exposed threads and the troubling disparallelism that makes the prosthetic treatment difficult, increasing the risk for biological complications and technical complications, respectively (Figure 5). Nevertheless, the patient refused a complete removable denture, so that implant removal would lead to placement of new implants, in combination with guided bone reconstruction and soft tissue management. This treatment plan may have the risks of implant failure and increased patient morbidity (Table 1 and Table 2). An implant-supported fixed dental prosthesis was initially excluded due to the implant’s disparallelism. Hence, a maxillary implant-supported overdenture was initially considered quite possibly the best therapeutic option.

Before definitive impression, multi-unit abutments were unscrewed, the implant connections were cleaned, and six OT Equator Titanium Abutments (Rhein’83, Bologna, Italy) with TiN coating were screwed (Figure 6), according to the manufacturer. Immediately after, the patient received a digital impression (CS 3600 intraoral scanner, Carestream Dental, Milan, Italy), taken at abutment level (Figure 7), using dedicated scan abutments (OT Equator Titanium Scan Abutment, Rhein’83). However, overlapping of the captured images failed to produce an accurate impression, maybe due to the disparallelism of the implant ranging from 65° to 86°. Hence, a prototype model was created and a convention gypsum impression [16] with a customized impression tray was taken at abutment level using conventional impression coping (Titanium Impression Coping, Rhein’83; Figure 8 and Figure 9). Esthetics, phonetics, the occlusal vertical dimension, and the centric relation were verified and approved by both the clinician and the patient (Figure 10 and Figure 11). Only after that, a definitive cast, implant position, and esthetic try-in were digitalized and a CAD/CAM titanium bar (NewAncorvis, Bologna, Italy) was anatomically designed with a dedicated software (Exocad DentalCAD, Exocad, Darmstadt, Germany) by a certified dental technician (MA), according to the prosthetic contours and the implant position. Three projects were created with a three-, two-, or one-piece CAD/CAM titanium bar, respectively (Figure 11b). Each of these projects failed to create an accurate titanium bar able to respect the minimum restorative space required by the prosthetic volume of the tried prosthesis, due to the multiple components involved (attachment system, prosthetic framework, artificial acrylic composite teeth, and pink acrylic base). Hence, a fourth project was created with a CAD/CAM titanium bar designed for a fixed screw-retained restoration, initially excluded due to the implant disparallelism. In order to overcome the undercuts created by the tilted implants, and to produce a passive-fit CAD/CAM titanium bar, abutments with an extragrade system were applied (Figure 11c). Extragrade is a special titanium abutment which has the possibility of entering, even in cases of strong disparallelisms, thanks to its internal design. On a large number of implants (those positioned worse from the point of view of the emergence profile), we cannot put the through screw. In this case, the abutment will be retained by the white Seeger only. The number of abutments without screws was limited to two, according to the manufacturer and considering the overall number of placed implants. Moreover, an angulated screw channel concept was applied to avoid the access hole in the vestibular area of the anterior teeth, moving the access holes in the pink area of the hybrid prosthesis. The fit of the implant bar was clinically and radiographically tested in the patient’s mouth, according to established criteria ([17,18,19]; Figure 12). It was important to position the flat surfaces of the abutment-bar connections in correspondence with the undercut created by the inclination of the implant; the flat surface indicates the location of the extragrade bevel, which allows the framework to overcome the undercuts created by the tilted implants. After that, an interocclusal record was taken in centric relation, and the prosthesis was delivered. The screw-retained fixed complete implant-supported prosthesis was seated in the patient’s mouth using the snap-on function offered served by the Seeger System (Figure 13). Four out of six screws were tightened according to the manufacturer, and the screw-holes were closed using composite pink materials. Finally, the occlusion was adjusted and radiographs were taken. The patient was scheduled for hygiene maintenance and control every 4 months (Figure 14, Figure 15 and Figure 16).

## 3. Discussion

A correct prosthetic plan and a prosthetic driven implant placement could help the dentist during surgery and the final rehabilitative steps. Moreover, a correct restorative-driven implant position offers important long-term advantages allowing for favorable esthetics and function, as well as optimal occlusion and implant loading [3]. This case report shows how incorrect initial planning of implant placement could determinate a final prosthetic problem and a consequent failure for the patient. In fact, improper implant positioning can result in esthetic, biological, and technical complications and can, in extreme situations, render the desired prosthetic rehabilitation impossible to achieve.

The patient of this case report, a partially edentulous 78-year-old woman with a screw-retained implant-support complete hybrid prosthesis in the upper jaw and natural dentition in the lower jaw was referred to a private center in Rome, Italy, due to several continuous breakages of the prosthetic part of the implant-supported rehabilitation. There are, therefore, two important parameters to consider: the emergence profile of the artificial tooth and the volume of hard and soft tissue that needs replacement [20].

The change in philosophy from “bone-driven” to “restoration-driven” implant dentistry was established with regard to the prosthetic reconstruction. The concept of virtual planning aims to optimize function and esthetics prior to implant placement [21]. In this context, computer-assisted implant-planning software has significantly improved and provided clinicians with excellent tools for pre-operative implant planning [22]. Careful and detailed treatment planning is enhanced [23], including the ability to allow immediate loading procedures [24,25].

The importance of correct implant and prosthetic planning and guided surgery is also to avoid problems related to the emergence profiles, the lack of prosthetic space, the impossibility of correct oral hygiene, and, not least, esthetic problems.

In this case report, after the first meeting with the patient, a fundamental decision to makefor the clinician was whether or not to remove or maintain previous dental implants. Some of the options evaluated would have been the removal of improper implants and placement of new implants, including guided bone regeneration (GBR), and further costs for the patients in terms of money and time. In fact, actual implants were placed too buccal, reducing the amount of bone available for new implants. To overcome this situation, a horizontal GBR would be needed to reconstruct the buccal bone and prevent peri-implant issues. Although GBR is a safe and predictable solution, complications may occur [26,27,28,29].

The clinician, after a second visit to the patient and accurate virtual planning, decided to maintain old implants and to solve only prosthetic problems. In the presented clinical case, the exposure of an implant surface could have represented a biological and esthetic problem. From an esthetic point of view, the titanium exposure creates a gray effect on the gum. Zirconia implants avoid this gray effect. Nevertheless, limited clinical evidence has been reported in the dental literature on their long-term survival and success rates. Moreover, a new biological abutment was used to replace the old multi-unit abutments and a new fixed hybrid prosthesis delivered to the patient.

To solve and overcome the incorrect implant placement and implant disparallelism, an OT Equator, in combination with a Seeger System and an extragrade concept, was used. The Seeger System of the OT Bridge allows you to have great stability, passivation and above all, manage the disparallelisms between the implants. This is because of the *Seeger* ring, which, thanks to its properties, manages to give stability and passivation, even in the presence of strong disparallelisms.

Virtual planning procedures simplify the decision-making process regarding the type of prosthesis and increase the predictability of esthetic and functional treatment outcomes [20]. The initial virtual idealized prosthetic set-up is crucial for the clinical assessment of a patient with an edentulous maxilla and is a requirement for proper computer-based virtual implant planning.

## 4. Conclusions

Although dental implants achieve high survival rates, the success of implant prosthetic therapy significantly relies on an appropriate implant position. Malpositioned implants can cause damage to vital structures, like nerves or vessels. Moreover, improper implant positioning can result in esthetic, biological, and technical complications and can, in extreme situations, render the desired prosthetic rehabilitation impossible to achieve. Low profile intermediate abutments with the elastic Seeger System, in combination with new digital technologies, seem to be a viable option for complicated prosthetic retreatment due to implant malpositioning.

## Figures and Tables

**Figure 1 materials-13-01883-f001:**
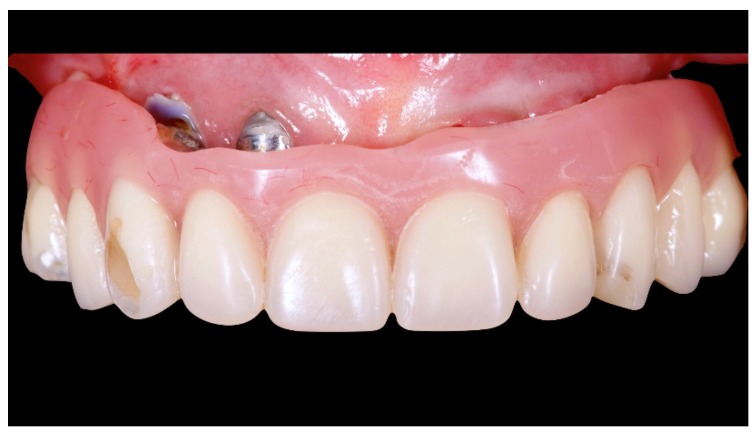
Picture of the patient’s work and initial situation.

**Figure 2 materials-13-01883-f002:**
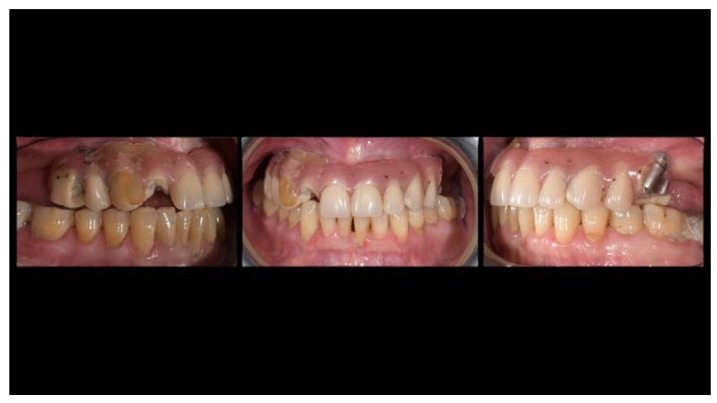
Pictures of detachments of tooth parts and pink resin from the prosthetic restoration.

**Figure 3 materials-13-01883-f003:**
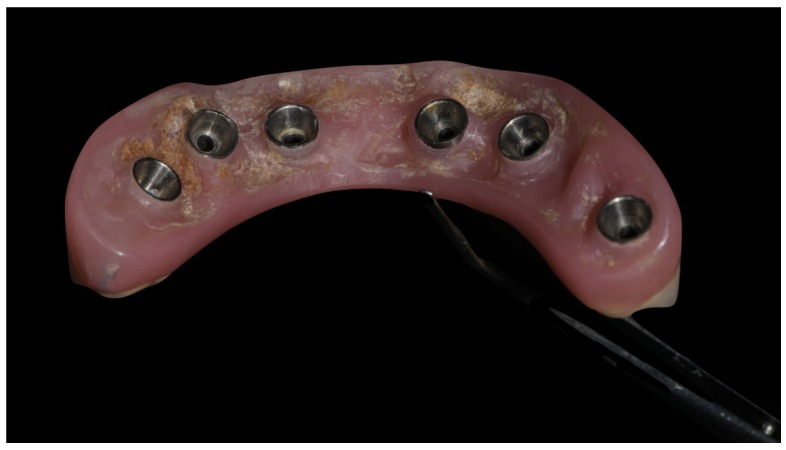
The internal part of the prosthetic restoration, the one in contact with the mucosa, shows the incorrect shape of the pink flange, which leads to the accumulation of plaque and food.

**Figure 4 materials-13-01883-f004:**
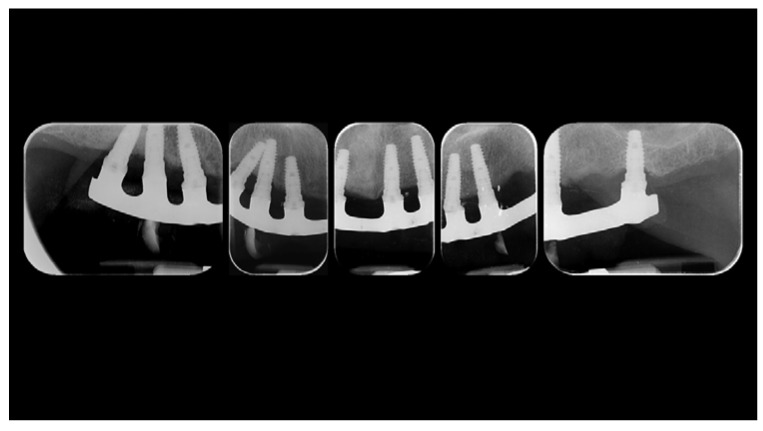
Radiographic status of the previous restoration.

**Figure 5 materials-13-01883-f005:**
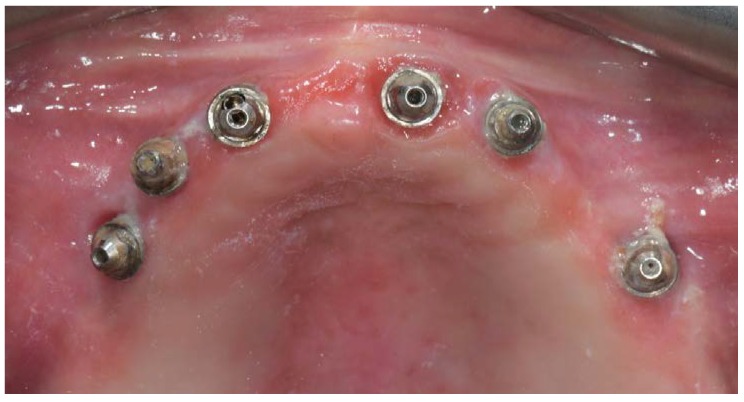
Intraoral picture of the implant positions and the multi-unit abutment screwed on.

**Figure 6 materials-13-01883-f006:**
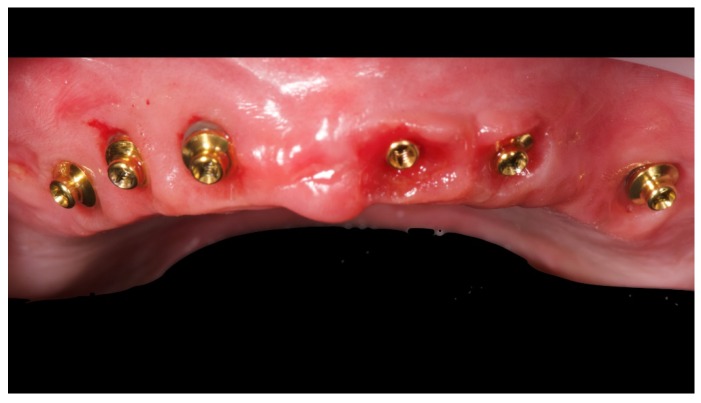
OT Equator used as abutment for the fixed prosthesis according to the OT Bridge system.

**Figure 7 materials-13-01883-f007:**
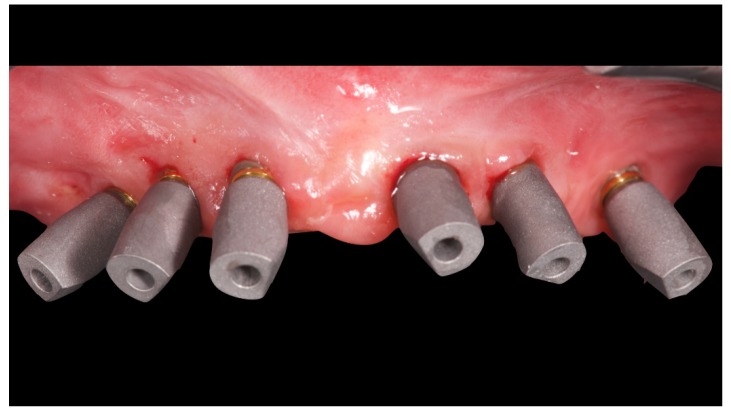
Scan bodies mounted on OT Equator. Note the strong disparallelism among the dental implants.

**Figure 8 materials-13-01883-f008:**
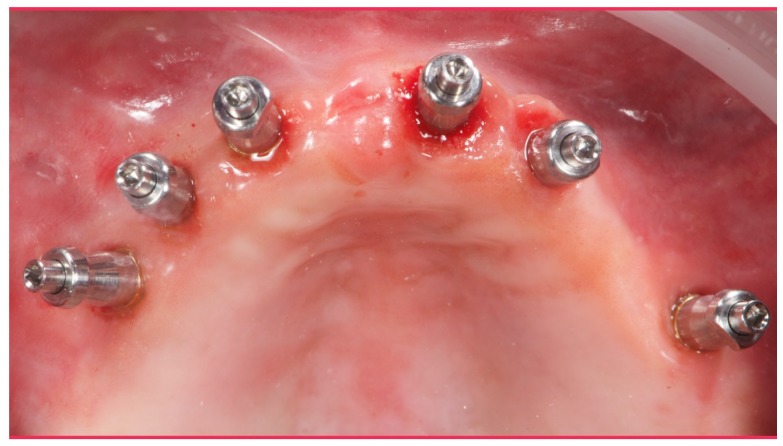
Analogical transfer screwed on OT Equator for classic impression technique.

**Figure 9 materials-13-01883-f009:**
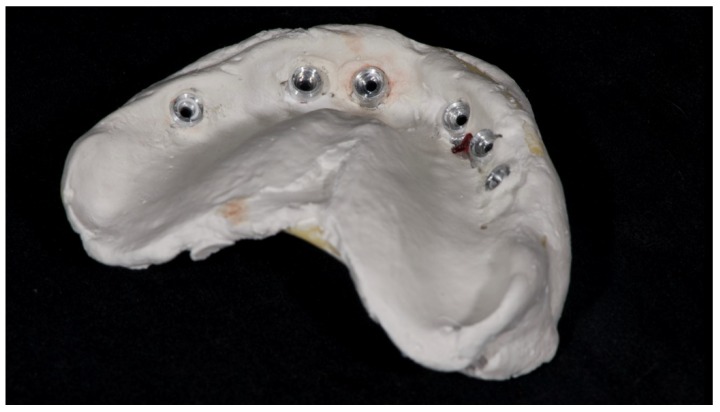
Internal detail of the plaster impression.

**Figure 10 materials-13-01883-f010:**
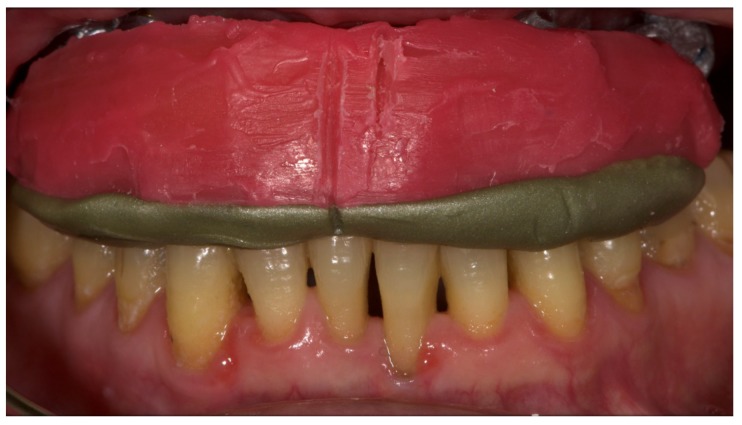
Vertical dimension of occlusion.

**Figure 11 materials-13-01883-f011:**
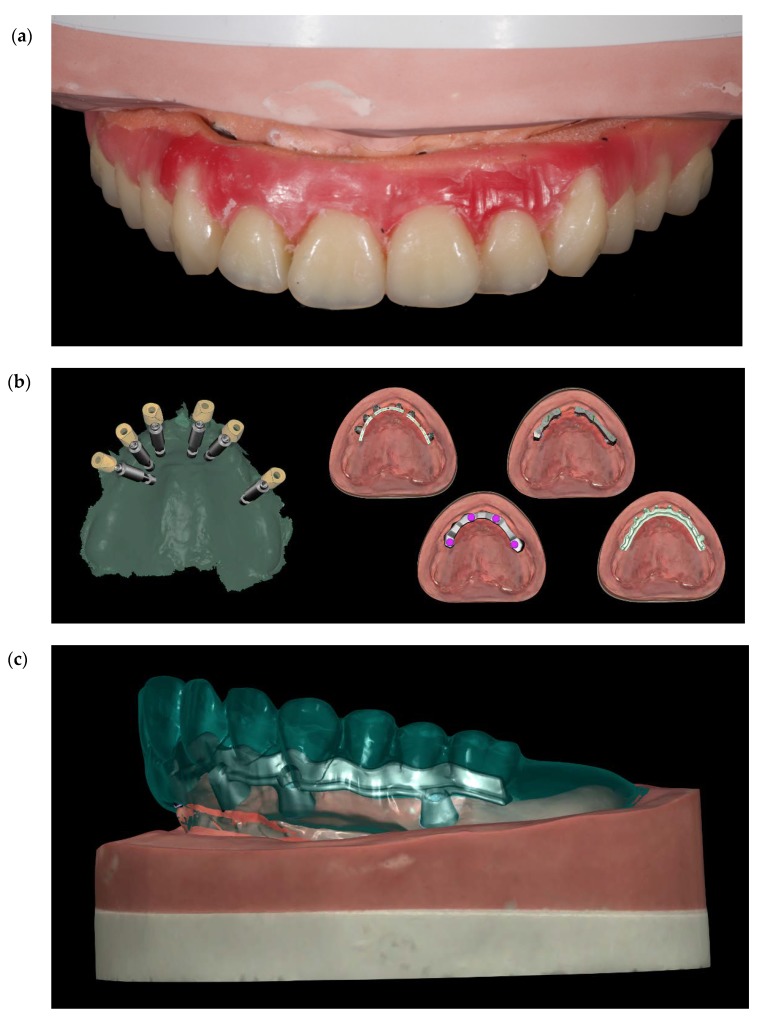
(**a**) Set-up of the teeth. (**b**) Four different CAD projects (**c**) Single frame titanium bar CAD project.

**Figure 12 materials-13-01883-f012:**
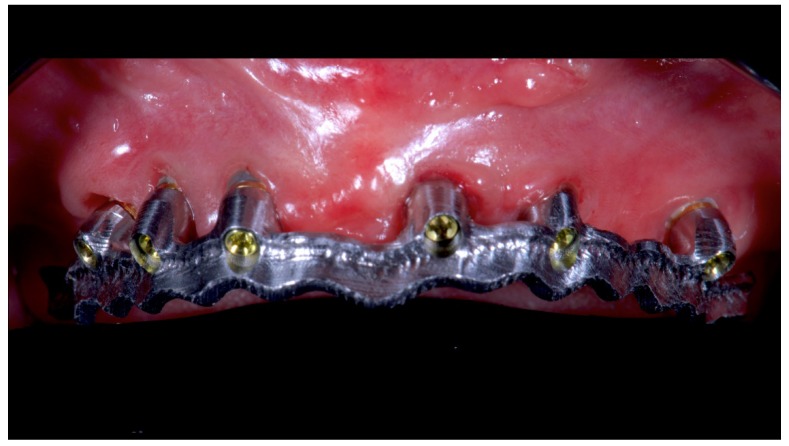
Single frame titanium bar mounted in maxillae with OT Bridge system.

**Figure 13 materials-13-01883-f013:**
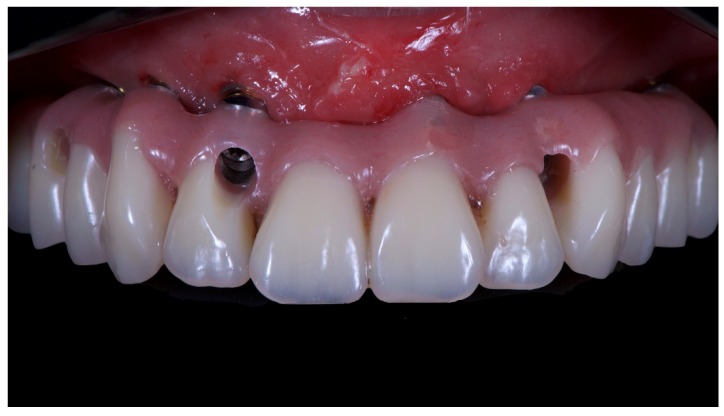
Final prosthesis mounted. Note the through holes. In two cases, it was compensated with only the *Seeger*, without a through screw.

**Figure 14 materials-13-01883-f014:**
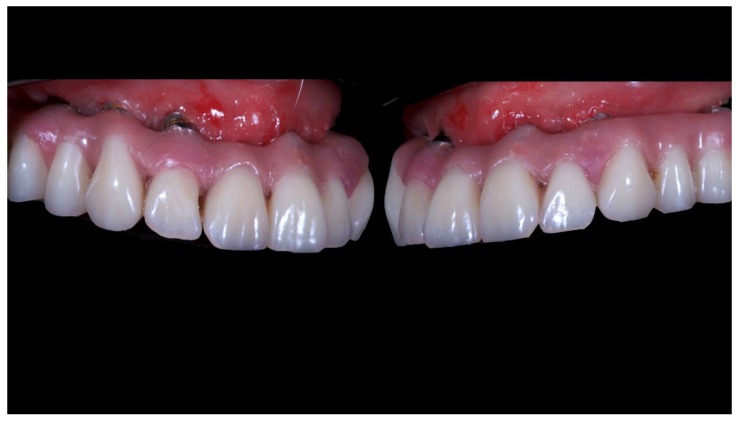
Final prosthesis with composite fillings made with pink and tooth-colored composite.

**Figure 15 materials-13-01883-f015:**
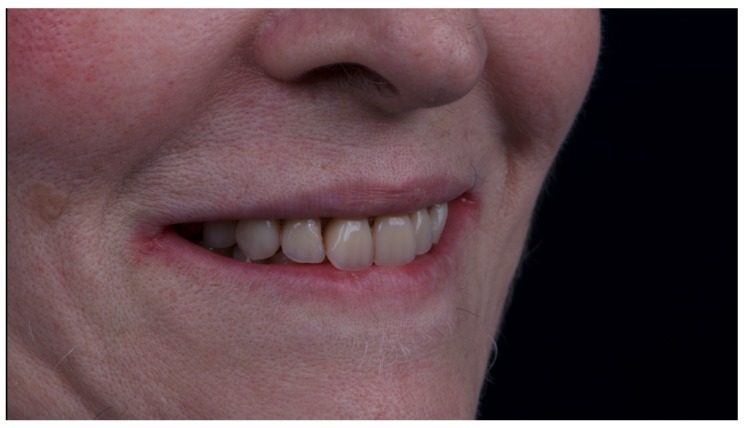
Smile test accepted by the patient.

**Figure 16 materials-13-01883-f016:**
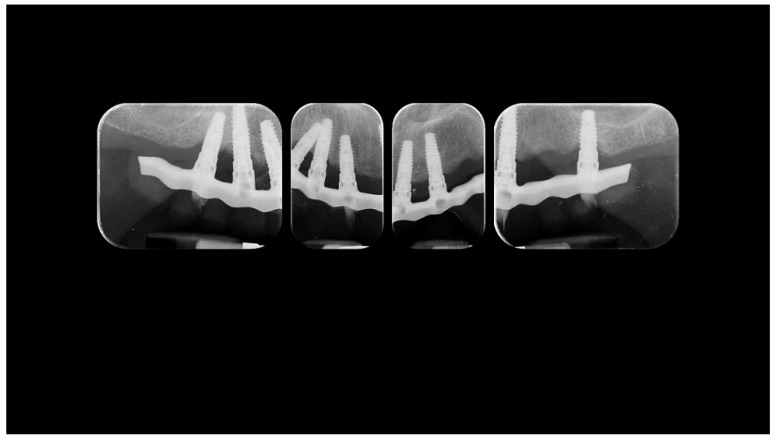
Radiographic status of the new work 1 year after delivery.

**Table 1 materials-13-01883-t001:** Implant retention.

Benefits	Risks
Minamally invasive approach	Peri-implantitis
Low cost for patient	Technical problems
	reliability

**Table 2 materials-13-01883-t002:** Implant removal.

Benefits	Risks
Accurate implant position	Needs for GBR
No surface exposure	Failure of the New Implants
	Surgical complications
	Higher cost

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
