# Peer review of "Errors in Implant Positioning Due to Lack of Planning: A Clinical Case Report of New Prosthetic Materials and Solutions"

_materials, 2020, doi:10.3390/ma13081883_

Round 1

Reviewer 1 Report

This article is a nicely detailed account of the use of state-of-the art implant materials and dental procedures for patient treatment. There are three general comments:

  1. Because the general readership of the journal Materials lacks a background in implant dentistry, it would be worthwhile at the outset to reference one or more of the well-known textbooks and/or review articles in this field.
  2. Because this journal has the focus of materials, there should be comments about why titanium or titanium alloy (Ti-6Al-4V) are used for the fabrication of dental implants, along with brief remarks about the clinical selection criteria for a particular implant product. The potential use of (hydroxyapatite) HA-coated titanium implants for appropriate clinical cases should be noted.
  3. While the manuscript is very clearly written, there are many places where relatively minor editorial corrections in the English are needed. The authors should work closely with the Editor to take care of this concern.

Author Response

Dear reviewers thanks to provide your feedbacks on this manuscript. I completely agree with you and the introduction and discussion sections have been improved according to your suggestions.

"Today, integrated treatment planning with dental implants represent a well established option [1,2]... 

...In the conventional protocol, titanium-based implants are usually placed freehand after implant sites preparation. Sand-blasted and acid-etched titanium implants are the gold standard in oral implantology, even if, the search on alternative materials and/or surface treatments is growing. The aims of these researches is to improve esthetics (ceramic implants) and osseointegration (hydroxyapatite coating or recently, hydrophilic surfaces)."

"In the presented clinical case, the exposure of implant surface could have represented a biological and aesthetic problem. From an aesthetic point of view, the titanium exposure creates a gray effect on the gum. Zirconia implants avoid this gray effects, nevertheless, limited clinical evidence has been reported in dental literature on their long term survival and success rates. "

English language has been revisited.

Reviewer 2 Report

The case report prepared by Authors seems to be interesting and valuable. It concerns the description of the prosthetic retreatment of an improper dental implant placement. Authors properly characterized the undertaken dental problem and subsequently proposed some treatment options. In general, the report is worth noticing but it requires some minor improvements – they are listed below.

  • In section Introduction Authors mentioned about some biological and technical complications that may be reduced by an ideal prosthesis. Please discuss this issue in more detail.
  • Considering the benefits and risks of the retention or the removal of implant such terms as “peri-implantitis” and “GBR” are used. These terms should be briefly characterized.
  • Authors proposed implants based on the titanium compounds – please briefly explain why exactly such implants were used instead of e.g. ceramic ones. Such an explanation has not been provided.
  • Final conclusions are too general and do not refer to the specific case that has been presented in paper therefore please conclude briefly the described treatment.
  • According to the requirements of the journal “In the text, reference numbers should be placed in square brackets (…)” therefore this should be corrected in the article because Authors use round brackets.

Author Response

Dear reviewer thanks for your appreciated feedbacks.

Introduction section has been improved with a paragraph about implant materials. Discussion has been improved focusing on problem with GBR. Conclusion has been improved too, adding primary conclusion form this case report.

Thanks to your feedbacks.

Reviewer 3 Report

I recommend the manuscript for publication in the "Materials". The manuscript described the clinical report in logical way and the method for treatment and the investigation were well described

Author Response

Dear reviewer thanks for your appreciated review and kindly comments.

Best regards.